# MSLF-Net: A Multi-Scale and Multi-Level Feature Fusion Net for Diabetic Retinopathy Segmentation

**DOI:** 10.3390/diagnostics12122918

**Published:** 2022-11-23

**Authors:** Haitao Yan, Jiexin Xie, Deliang Zhu, Lukuan Jia, Shijie Guo

**Affiliations:** 1Academy for Engineering & Technology, Fudan University, Shanghai 200433, China; 2School of Mechanical Engineering, Hebei University of Technology, Tianjin 300132, China

**Keywords:** diabetic retinopathy, image segmentation, multi-scale, multi-level, feature fusion

## Abstract

Diabetic Retinopathy (DR) is a diabetic complication that predisposes patients to visual impairments that could lead to blindness. Lesion segmentation using deep learning algorithms is an effective measure to screen and prevent early DR. However, there are several types of DR with varying sizes and high inter-class similarity, making segmentation difficult. In this paper, we propose a supervised segmentation method (MSLF-Net) based on multi-scale–multi-level feature fusion to achieve accurate end-to-end DR lesion segmentation. MSLF-Net builds a Multi-Scale Feature Extraction (MSFE) module to extract multi-scale information and provide more comprehensive features for segmentation. This paper further introduces the Multi-Level Feature Fusion (MLFF) module to improve feature fusion using a cross-layer structure. This structure only fuses low- and high-level features of the same class based on category supervision, avoiding feature contamination. Moreover, this paper produces additional masked images for the dataset and performs image enhancement operations to ensure that the proposed method is trainable and functional on small datasets. The extensive experiments are conducted on public datasets IDRID and e_ophtha. The results showed that our proposed feature enhancement method can perform feature fusion more effectively. Therefore, In the end-to-end DR segmentation neural network model, MSLF Net is superior to other similar models in segmentation, and can effectively improve the DR lesion segmentation performance.

## 1. Introduction

Diabetic Retinopathy (DR) is one of the most common retinal vascular complications of diabetes mellitus, mainly manifested by high blood sugar that causes insidious and uninterrupted damage to the retinal blood vessels [1]. DR could eventually lead to vision impairments that could potentially reach blindness in patients. However, DR has no obvious symptoms in its early stage and is not easily diagnosed. Once it progresses to an advanced stage, the condition becomes irreversible [2]. Many patients have already missed the best time for treatment by the time they discovered their impaired vision. In developing countries, the incidence of diabetic retinopathy has become the second leading cause of low vision or blindness in the adult working population after hereditary [3].

Ophthalmologists usually look for DR lesions in fundus images to diagnose the condition. In clinical practice, there are four types of abnormal lesions in DR: Micoraneurysms (MA), Hemorrhage (HE), Soft Exudate (SE), and Hard Exudate (EX) [4]. Doctors can view the type, size, distribution, and number of the four lesions via manual segmentation to determine the disease’s progression [5] (see Figure 1). However, the high cost and limited physician resources have made it difficult for the manually segmented DR screening technology to become widespread on a large scale [6]. With technology development, automatic lesion segmentation technology has gradually matured and been applied in clinical diagnosis.

The mainstream automatic DR segmentation techniques are divided into two types: non-deep and deep learning methods [7]. The prior methods rely on manual feature extraction, mainly morphological processing, classification, and region-growing methods [8,9,10]. However, these segmentation methods are not highly accurate; they also rely on manual feature extraction by professional doctors [11]. Therefore, it is difficult to employ them in medical diagnoses. Deep learning methods use automatic feature extraction and skip the traditional feature engineering steps [12]. This enables an end-to-end working approach and reduces the need for domain expertise. Fully convolutional neural networks (FCN) were the first deep learning methods applied for this purpose [13]. Subsequently, the emergence of U-Net pushed deep learning methods to their pinnacle [14].

Particularly, Noushin Eftekhari et al. [15] proposed a convolutional neural network-based method for MA single lesion segmentation. They used a two-stage training strategy to first select lesion candidates. They then used Convolutional Neural Networks (CNN) to classify MA and non-MA. However, their approach was time-consuming and labor-intensive. Moreover, the segmentation accuracy was not adequately high. Mo et al. [16] designed a Fully Convolutional Residual Network (FCRN) incorporating multi-level information for HE lesion segmentation. The FCRN could quickly and accurately segment lesions, avoiding extensive pre- or post-processing steps. However, the FCRN is only effective for EX lesions and not for the other three types of lesion segmentation. Li et al. [17] produced the first private dataset DDR for multi-lesion segmentation and attempted multi-lesion segmentation tasks using PSPNet [18] and DeeplabV3+ [19]. Guo et al. [20] developed an end-to-end multi-lesion segmentation network called L-seg, which is FCN-based. They also introduced a weighted fusion module and achieved promising results.

Although the neural network-based DR segmentation methods have achieved certain results, DR segmentation is still a challenging task due to the following reasons. First, four types of lesions exist in DR; they have high similarity between lesion classes and large intra-class distinctions. Moreover, the lesion sizes are extremely small and their shapes greatly vary; this makes effective feature extraction difficult. Second, the retinal structure is complex; it has anatomical structures such as blood vessels and optic discs, which are easily confused with lesions. Moreover, the retina’s size is much larger than that of DR lesions. It is a great challenge to segment small lesions with complex structures in retinal images. Third, medical images often involve patient privacy and high-cost manual annotation, resulting in a great lack of available public datasets. The quality of the available datasets is not ideal for adequate model training. Overall, it is necessary to further investigate the four lesion features of DR and propose a DR multi-lesion segmentation network with a wider application.

In this paper, we propose a Multi-Scale–Multi-Level feature (MSLF) fusion model to conduct the problem of DR segmentation. The proposed MSLF-Net contains a Multi-Scale Feature Extraction (MSFE) module and a Multi-Level Feature Fusion (MLFF) module. The former builds a pyramidal feature extraction block, which is used to extract multi-scale information and provide more comprehensive features for segmentation networks, and the latter facilitates multi-level feature learning of the same category by a cross-layer structure that allows lower-level features of the same category to participate and enhance the semantic classification of higher-level features. MSLF-Net effectively improves the model’s segmentation accuracy for DR using the above improvements and outperforms other comparable end-to-end models in segmentation.This paper’s innovations and research contributions are summarized as:1.This paper established effective pyramid feature extraction blocks, which can generate multi-scale features efficiently and enhance the segmentation of small lesions significantly.2.This paper proposed the cross-layer and multi-scale class response feature fusion method, which makes the multi-scale feature fusion more effective, can adequately meet the challenges of small or complex features, avoids the complicated preprocessing work, achieves a high-performance end-to-end segmentation network, and reaches an advanced level in DR lesion segmentation.3.This paper improved the image quality by preprocessing the field of view (FOV) mask images for the IDRID and e_ophtha dataset, so that the model can be trained and worked on small datasets, and to facilitate the future work of DR lesion segmentation.

The rest of this article is organized as follows. Section 2 elaborates the data and methodological concepts employed in this work. Section 3 records and describes the experimental results. Then, Section 4 discusses the experimental results. Finally, Section 5 concludes the paper.

## 2. Related Work

### 2.1. Convolutional Neural Networks in DR Lesion Segmentations

CNN is widely used in medical image segmentation because it can automatically extract the most appropriate features, which improves CNN’s segmentation performance and reduces the need for professional knowledge in traditional feature extraction.

The FCN network is the first convolution neural network applied to medical image segmentation (realizing the pixel-by-pixel image segmentation task for the first time). Tan et al. [21] designed a network with 10 convolution layers to segment MA, HE, and EX, but the network had too few convolution layers to extract the advanced features of lesions, and the overall segmentation accuracy was not high. Kou et al. [22] proposed a deep recursive convolution neural network, which combines residual structure with recursive convolution operation and applies it to the convolution layer. It can accumulate more effective features than typical CNNs and achieve better performance in MA segmentation tasks, but the method has a high miss detection rate. Huang et al. [23] designed a Transformer-based Relation Transformer Network (RTNet) for DR lesion segmentation. RTNet first uses a convolution network to extract DR lesion information and retinal vessel information, and then uses the cross-attention mechanism to obtain the distribution relationship between DR lesions and retinal vessels. By pathological dependence of lesions and blood vessels, the possibility of network mis-segmentation is reduced, and the highest performance is achieved in DR lesion segmentation tasks. However, RTNet needs to be divided into two phases during training, which is complex and too dependent on external information.

CNN has achieved great success in DR segmentation, however, it also faces new difficulties. On the one hand, the size of DR lesions is very small, some lesion areas are only a few pixels in size, and the CNN down-sampling operation can easily result in the loss of information about minor lesions. On the other hand, DR lesions often have extreme shapes (such as slender, narrow, and high), which make it difficult to extract features from CNN. This also limits the application of the pure CNN network in DR segmentation.

### 2.2. Feature Fusion in Medical Image

To solve the problems faced by the pure CNN network in extracting medical images such as DR lesions, researchers have proposed many improved methods, among which feature fusion effectively improves the segmentation performance of CNN on medical images. According to the operation mode, common feature fusion methods include element-wise product, feature concatenation, parallel fusion of two models, and improved feature concatenation based on the attention mechanism or deconvolution, etc. U-Net is the improved feature concatenation fusion network based on skip, which combines the shallow features in the encoder with the deep features in the decoder through the skip structure. This enables U-Net to utilize both the high-resolution of the shallow features and the high-semantic information of the deep features, and has a good segmentation effect for organs or lesions with multiple scales and large morphological differences. Zhang et al. [24] uses the attention mechanism to regulate feature concatenation fusion at different semantic levels, which is superior to U-Net in detecting MA lesions. However, there is a large semantic level difference between the shallow features and the deep features, which makes it difficult for them to merge effectively, and feature pollution may even occur [25]. This makes the feature fusion method not effective for the segmentation of complex lesions. DeepFusion, proposed by Song et al. [26], provides a fuller and more comprehensive feature fusion. DeepFusion has two data input ports, one for the main data and the other for the associated data. Therefore, DeepFusion not only extracts the global and local features of information, but also fuses the features of the two datasets. Finally, DeepFusion fuses the global and local features through element-wise product, and greatly improves the accuracy and efficiency of feature extraction through two feature fusions. However, the data types used by DeepFusion differ from medical image segmentation data, making it more difficult to obtain the related information of medical image segmentation datasets. Especially for the DR lesion segmentation, there is only a single lesion segmentation dataset, and there is no corresponding associated information for the research. Ramamurthy et al. [27] designed a two-model parallel structure. One model extracts the prominent features of lesions, and the other model calibrates the features effectively. By fusing the two parallel features of models, we obtain more accurate high-level spatial information. This method has some advantages in identifying morphologically similar lesions, but it does not solve the problem (loss of micro features) caused by pooling and strided convolution.

## 3. Materials and Methods

In this section, the datasets and data processing methods are first introduced. Then, we give a brief overview of our proposed model, and elaborate on the components of the model. Finally, The methods for evaluating the performance of models are described.

### 3.1. Datasets

The proposed model MSLF-Net is evaluated on two public datasets, including IDRID [28] and e_ophtha [29]. The IDRID dataset comes from the 2018 Diabetic Retinopathy Segmentation and Grading Challenge, which is the only dataset with four kinds of DR lesions. IDRID consists of 81 cooler fundus images and lesion annotations with resolution of 4288 × 2848. In the experiment, data were divided into training and testing sets, 54 images for training and 27 images for testing. e_ophtha provides only two types of lesion annotations, including EX lesions and MA lesions. Its images have various resolutions such as 2544 × 1696, 1440 × 960, etc. e_ophtha_EX consists of 47 color fundus images, and we set 32 images for training and 15 for testing. e_ophtha_MA has 148 images. The training set and testing set are 20 and 8, respectively.

### 3.2. Image Preprocessing and Augmentation

Image preprocessing and augmentation aim to resize the dataset images and increase image count. The images of two datasets used in the paper had high resolution and several pixel sizes, which makes it difficult to use them as input in our model. Accordingly, we resized the images to 1440 × 960 pixels without deformation. To conduct model training on a big dataset, images were additionally augmented to enhance the dataset. We performed five random geometric transformations with a 50% probability: random vertical flip, random horizontal flip, random affine, random rotation, and random adjustments of brightness, contrast, saturation, hue in the range of 0.07.

In addition, only the circular area of the eyeball in the fundus image contains valid information, and the rest are filled with black pixels. In order to highlight the circular area of the eyeball and avoid irrelevant black pixels from affecting the network training effect, we imitated the fundus retinal datasets such as DRIVE [30] to make mask images for each image of IDRID and e_ophtha. The image of IDRID, lesion annotation, and mask are shown in Figure 2.

### 3.3. Model Architecture

#### 3.3.1. Model Overall

The structure of MSLF-Net was inspired by U-Net and adopts a symmetric encoder–decoder structure. The encoder part uses the pre-trained Vgg16 [31] to enhance the feature learning capability. The decoder uses transposed convolutional up-sampling. Such structure allows for effective feature integration. The model adds MFEF and MLFF to U-Net. The former fully extracts multi-scale information to enhance the model’s feature extraction capability. The latter enhances network segmentation accuracy through a cross-layer architecture that allows both lower- and higher-level features to participate in semantic classification. These two modules complement each other to achieve multi-scale feature fusion on the feature map of the same activation class. This ensures that the model has a stronger discriminative ability for lesions with variable scales and complex features. Moreover, it could obtain a finer segmentation capability. Additionally, we adopted the idea of deep supervision strategy [32] to retain the original output layer in the U-Net decoder, which is used to suppress background pixels and provide detailed lesion localization and structural information to the model.The overall structure of MSLF-Net is shown in Figure 3.

#### 3.3.2. Multiscale Feature Module (MFEF)

The MFEF module’s purpose is to leverage the physical contour feature information from the network’s shallow layer and the deep semantic category information from its deep layer. In general, the network’s shallow layer captures low-level information, such as points, lines, edges, and textures, while the deep layer extracts semantic category information. Additionally, it has a small down-sampling multiple and high resolution, which is suitable for segmenting small lesions. On the other hand, the deep layer does the opposite. There are lesions of different sizes in the fundus images of DR. Predictions using a single resolution image are ineffective. We borrowed the FPN [33] idea and designed a pyramidal multi-scale feature extraction module to collect the feature map information with different resolutions. To retain the extracted multi-scale feature information as much as possible, the MFEF module was designed with activation and convolution operations based on the idea of ResNet [34], and the “add” method was used to fuse the convolved feature maps before and after the operations.

The structure is shown in Figure 4. We passed each of the decoder module’s five convolution blocks through the MSFE module to obtain five new convolution blocks. The first four convolution blocks need to be up-sampled to the original map’s size via bilinear interpolation. The last convolutional block does not need to be up-sampled because its size is identical to that of the original one.

#### 3.3.3. Feature Fusion Module (MLFF)

The specific operation of the MLFF module is to improve feature concatenate through feature channel rearrangement. The MLFF module was designed to avoid a brutal approach to directly fusing the multi-scale features extracted by the MSFE module, which can fully utilize the multi-scale information in a manner ensuring that the same-class activation features are fused.

In convolutional neural networks, there are two common methods of feature fusion. One operation is to add the feature maps element-by-element, such as Resnet, SSD [35], etc. The other operation is to add the feature maps by dimension and then adjust the number of channels with convolution, such as Densenet [36], U-Net, etc; this is shown in Figure 5a,b.

The above two fusion methods are only suitable for shallow feature fusion but not for deep feature fusion with existing category attributes. For image segmentation tasks, the label map is processed by One-hot coding, and the category information is sequentially mapped to different channels using a serial number. Then, the supervised output layer’s channels are also sequentially activated by category. If these feature maps are directly summed or processed with splicing operations, the activated channel category information will be destroyed. To protect each channel’s category information, the MLFF module performs a slice, splicing across the convolutional layers, instead of the corresponding regular splicing in the convolution. We wish to constrain each channel of these multi-scale feature maps implicitly through the output layer to carry the most relevant information to the corresponding category. This guides the feature maps extracted by the MFEF module to converge to a category identity.

As shown in Figure 5c, the same color images represent feature maps containing information of the same semantic category. The MLFF module classifies the channels of the convolution blocks by semantic information. The number of channels in each convolution block is identical to the number of categories, while each channel corresponds to the category activation of a class. In the feature fusion operation, the MLFF module stitches the channels of different convolutional blocks with the same category, according to the “Concatenate” connection method to highlight the information of different scales in the same category. The specific operations are as follows:

As shown in Figure 3, the MSFE module generates five convolutional blocks, each with c channels: (1)$$A_{o}=\left(A_{\left(c\right)}^{1},A_{\left(c\right)}^{2},A_{\left(c\right)}^{3},A_{\left(c\right)}^{4},A_{\left(c\right)}^{5}\right)$$

Then, MLFF spliced the channels of the same category through a slice operation across convolutional blocks to generate $A_{f}$: (2)$$A_{f}=\left(A_{\left(1\right)}^{1},A_{\left(1\right)}^{2},A_{\left(1\right)}^{3},A_{\left(1\right)}^{4},A_{\left(1\right)}^{5},\ldots \ldots ,A_{\left(c\right)}^{1},A_{\left(c\right)}^{2},A_{\left(c\right)}^{3},A_{\left(c\right)}^{4},A_{\left(c\right)}^{5}\right)$$

Finally, c-group 1*1 convolution is performed on $A_{f}$ to get the output layer as shown in Figure 5c: (3)$$Output=(A_{\left(1\right)},A_{\left(2\right)},A_{\left(\ldots \right)},A_{\left(c\right)})$$

### 3.4. Implementation Details and Experiment Settings

In MSLF-Net, the encoder consists of five convolutional blocks, while the decoder is composed of one transition block and five up-sampling blocks. The five convolution blocks in the encoder have the same structure as Vgg16 and use its pre-training weights on the ImageNet dataset [37] to initialize. The structure of the decoder is the same as that of U-Net. The MFEF module compresses the channel of the decoder convolutional blocks to five. While the MLFF module rearranges and compresses the convolutional blocks processed by MFEF module according to the channel order to obtain the output. The detailed configuration of MSLF-Net is shown in Table 1.

The loss function is a hybrid loss function, including the sweighted cross-entropy and dice loss functions. The loss function formula is as follows:

The weighted cross-entropy loss function formula is: (4)$$L_{cross}=-\sum_{c=1}^{M}\omega_{c}y_{c}log\left(p_{c}\right)$$
where, *M* (*M* = 5) and *c* are the numbers of class and class, respectively. ω stands for class-weight, and the value is set as 100, 100, 50, 30, 1 for MA, HE, SE, EX, and background, respectively, which is calculated by the ratio of the number of labeled pixels to the number of all pixels in the dataset. *y* denotes the label value, and *p* indicates the predicted probability.

The dice loss function formula is: (5)$$L_{dice}=c-\sum_{c=1}^{M}\frac{2*TP_{p}\left(c\right)}{2*TP_{p}\left(c\right)+FN_{p}\left(c\right)+FP_{p}\left(c\right)}$$
where, $TP_{p}\left(c\right)$, $FN_{p}\left(c\right)$, and $FP_{p}\left(c\right)$ are the predicted probabilities of true positive, false negative, and false positive for each class, respectively; *c* and *M* have the same meaning as the above formula.

The hybrid loss function is: (6)$$L_{loss}=L_{cross}+\lambda *L_{dice}$$
where λ is the weight ratio of two loss functions, which is set to 1.0. The influence of λ on the segmentation performance of the model will be discussed in the ablation experiment.

We implemented the experimental architecture using PyTorch, and executed it on NVIDIA 3090 GPU. In the experiment, we trained models for 200 epochs using the SGD optimizer with momentum 0.9 and weight decay 0.0005. We adjusted batchsize to 2 and Initial learning rate to 0.01. In addition, we adopted the Poly strategy to dynamically adjust the learning rate during training. The calculation formula of the Poly strategy is as follows: (7)$$lr=Initial\_lr\times {\left(1-\frac{epoch}{total\_epoch}\right)}^{0.9}$$

The epoch-wise learning rate curve and loss curve during training on the IDRID dataset are presented in Figure 6.

### 3.5. Model Performance Evaluation

The number of background pixels in DR images is much larger than the number of lesion pixels. For high class-imbalance, the PR Curve is concerned with True-Positive predictions. Therefore, the PR curve is generally considered to be better than other indicators in this situation. To compare the segmentation performance of different models visually, AUPR is also introduced. AUPR is the value of the area under the PR curve. The closer the AUPR value is to 1, the better the model’s predictive performance. In addition, we calculated the mean value from 5 experiments as the final experimental result to evaluate the overall model performance for multi-lesion segmentation of DR.

## 4. Results and Discussion

In this sub-section, we present the results of comparative experiments and ablation experiments, the former is used to evaluate the segmentation performance of MSLF-Net, and the latter revealed distinct functions associated with different modules in MSLF-Net.

### 4.1. Comparative Experiments

We compared our method’s segmentation performance with publicly available previous works on the IDRID dataset. The methods involved in our experiments included FRCN, UNet++ [38], MANet [39], PSPNet, DeeplabV3+, L-Seg, and RTNet. Note that RTNet has additionally introduced the retinal blood vessel dataset. We need to remove its blood vessel information and compare it with the results of RTNet-base. In addition, we introduce the feature fusion method proposed by Ramamurthy et al. [27]. The following experimental results are taken from the average of five experiments.

As shown in Table 2, our method ranked first in the AUC_PR of SE, outperforming other compared methods. It ranked second in the AUC_PR of MA, EX and HE, showing a slight disadvantage. It also achieved the highest score in the mAUC_PR metric. To sum up, Our method has achieved an advanced performance compared to other methods in segmentation of four lesions. The method of L-Seg outperformed our method in MA segmentation. This is probably because our proposed multi-scale feature extraction strategy loses some features, rendering our boosting effect to be not as good as the multi-level side supervision approach used by the L-Seg. RTNet-base propose a transformer-based attention block, which obtained detailed lesion information. RTNet-base has achieved the best results in the segmentation of HE and EX lesions. However, the transformer block pays more attention to the global relationship and ignores the local relationship, which seriously affects the segmentation effect of SE lesions. The feature fusion method proposed by Ramamurthy et al. still has problems of feature loss caused by pooling and strided convolution, so it does not perform well in DR segmentation.

The PR curve obtained from the experiment is shown in Figure 7. The higher the PR curve, the larger the area included, indicating a better performance of the model.

We also validated our proposed method’s effectiveness and reliability on the e_ophtha dataset. Compared with the IDRID dataset, the e_ophtha had only two DR lesion annotations, MA and EX, while it had lower image resolutions and a large number of low-quality images with issues of uneven lighting, underexposure, overexposure, etc. Performing segmentation experiments on the e_ophtha dataset was more challenging and placed higher demands on the model. As shown in Table 3, compared with other methods, our method achieved the best results for the segmentation of both lesions. We could observe that compared with the segmentation results on the IDRID dataset, the segmentation results of all the methods on the e_ophtha dataset showed different degradation degrees. Especially in the segmentation results of MA lesions, the segmentation performance of other methods decreased by 54% to 89%, respectively; however, our method’s segmentation performance decreased by 49%, showing the strongest robustness among all methods. Generally, our method performed poorly in segmentation for MA because, on one hand, the MA lesion size was too small and had only a few pixels in size, which made feature extraction difficult. On the other, the retinal vessels were similar to the MA lesion features, causing a high occurrence of missegmentation phenomena. The PR curve obtained from the experiment is shown in Figure 8.

To sum up, the results presented show that Our 3M-Net outperforms the other five methods.

### 4.2. Ablation Studies on IDRID Dataset

#### 4.2.1. Analysis of the Model Components

In this sub-section, we also performed ablation experiments to further understand the impact of each component on the model. First, we analyzed the effect of multi-scale information with the baseline model. Then, we discussed how the general fusion and cross-layer category fusion approaches helped the model. From Table 4, it is clear that these two modules effectively improved its segmentation performance. We introduced the MFEF module to provide the model with complete multi-scale information. The introduced MLFF module integrated the corresponding category-specific activations from different scales as the final fusion through implicit supervision of the final output. This fusion method effectively avoids the mutual interference of category information and feature contamination during feature fusion.

#### 4.2.2. Analysis of the Hybrid Loss Function

The loss function used in this paper consists of two parts. In order to verify the influence of loss functions with different weights on segmentation performance, we choose the appropriate λ in Formula (6) through ablation experiments. Where, λ = −1 means only DICE loss function works, and λ = 0 means only cross-entropy loss function works. With the increase of λ, the weight of the Dice loss function becomes larger. Table 5 shows that the segmentation performance of the model is poor when using a separate loss function. When using the mixed loss function, the segmentation performance of the model is significantly improved. With the increase of λ, when λ = 1.0, the segmentation performance of the model was the best. When λ continued to increase, the segmentation performance of the model started to decline. Therefore, the optimal λ is 1.0.

#### 4.2.3. Analysis of the Image Augmentation

In view of the shortcomings of DR in segmenting datasets, We conducted image preprocessing and Augmentation for the IDRID and e_ophtha datasets. Table 6 shows the ablation experiments of FOV mask image and geometric transformations. The experiments show that both can significantly promote the segmentation performance of the model, especially with the increase of FOV mask image, the mAUPR index of the model increases by 9.7%.

### 4.3. Discussion

There are several methods of examining fundus images, including ophthalmoscopy, optical coherence tomography (OCT), fluorescein fundus angiography (FFA), and fundus color photography [40]. The first three methods either require high-cost equipment or some clinically restricted agents, such as pupil diluents and contrast agents, making it difficult to popularize their use. In contrast, fundus images are not harmful and can be digitally stored, making them suitable for large-scale lesion screening. This study’s main contribution is the lesion segmentation of fundus images by the neural network to assist doctors in diagnosing DR conditions. Presently, there is still much room for improvement in DR segmentation techniques due to the complexity of fundus images and DR lesions [41]. These issues motivate us to further investigate this problem.

The purpose of DR segmentation is to accurately mark all lesion regions. The main problems faced in this process are the difficulties of tiny feature extraction and noise interference [42]. To address these issues, we proposed a multi-scale cross-layer feature fusion strategy and conducted comparative experiments with common feature enhancement methods, such as pyramid pooling, dilated convolution, feature superposition, attention mechanisms, and depth supervision. The experimental results demonstrated that our proposed method adequately performed the DR segmentation tasks, and our depth-supervised approach also had merits in tiny features extraction.

The main problem we faced on the e_ophtha dataset was noise interference. The large number of low-quality images made it difficult for our model to extract features. Thus, its segmentation performance for MA substantially deteriorated. As shown in Figure 9, in the segmentation result graph of MA, each method showed missed segmentation. Particularly speaking, UNet++ had the most serious problems, while DeepLabV3+ and our proposed model maintained correct segmentation for larger lesions. These results suggested that null convolution is a method that can be learned from or adopted in future studies.

For the IDRID dataset, the segmentation performance of HE, SE, and EX was satisfactory, but the segmentation performance for MA lesions with smaller sizes was not. This dataset’s image quality was high, such that feature extraction was relatively simple. The main problem we faced was with tiny feature extraction. Figure 10 shows the four results of lesion segmentation plotted for the IDRID dataset.

With a comparison between the segmentation result plot of each method in Figure 10 and GroundTruth, it is apparent that all methods did not have any missed segmentation of lesions. However, missegmentation occurred frequently; a large number of normal retinal tissues were missegmented as lesions. From this phenomenon, we conclude that distinguishing between normal retinal tissues and lesions is the key to improving segmentation performance in the presence of high-quality fundus images. The occurrence of DR is associated with structural changes in the retinal vasculature. For example, MA is an abnormal proliferation of retinal capillaries following swelling, and HE is a progressive lesion resulting from the rupture of a retinal vessel or microaneurysm. Therefore, the color, contour, and other appearance features of many early lesions are extremely similar to those of blood vessels. This fact is the main reason for missegmentation. However, we can obtain useful information from this pathological connection. We carefully analyzed the pathological relationship and combined it with the observation of the multi-lesion segmentation results in Figure 10. The distribution of DR lesions is related to the distribution of retinal vessels. The existing methods have ignored this pathological distribution relationship, resulting in a large number of missegmentations that could have been avoided. This topic is what we need to focus our attention on in future studies.

Overall, as shown in the segmentation results in Figure 9 and Figure 10, our proposed method significantly improves the DR segmentation accuracy. This result is valuable both for enhancing physicians’ confidence in automatic DR segmentation techniques and for screening the conditions of general diabetic patients. Our method accelerates the application of deep-learning algorithms to DR clinical diagnosis to a certain extent, partially overcoming the problems of physician shortages and high work intensity. These achievements are prerequisites for moving an approach from theory to clinical practice. However, neural networks are a data-driven science. Available DR segmentation datasets are not very abundant, in which there are insufficient data samples and solidified characteristics by the devices and patient distribution areas where the samples are collected. We are currently inconclusive about the existence of data bias in geographical distributions. However, the inadequate training of models, a limited generalization ability, and the lack of reproducibility arising from data bias and insufficient datasets are obvious issues. They are evidenced by the fact that the accuracy and generalization of the DR segmentation techniques are much lower than those of other medical image segmentation approaches. This phenomenon severely hinders clinical applications and is currently a pressing problem.

## 5. Conclusions

In this paper, we proposed a new automatic DR lesion segmentation model, MSLF-Net. This model could fully extract multi-scale information and perform effective class hierarchical feature fusion to improve the segmentation accuracy of DR lesions via feature enrichment and feature contamination avoidance. It also has an enhanced anti-noise capability. Moreover, we optimized the DR segmentation dataset to improve the model’s training efficiency and quality. To verify the validity and reliability of MSLF-Net, we conducted comparative experiments on the IDRID and e_ophtha datasets. Then, we conducted ablation experiments for each component of the model to understand its impact. The experimental results showed that our model has a satisfactory performance in DR segmentation compared with the existing DR segmentation methods. This study can facilitate large-scale screening and condition diagnosis of DR. In the future, we plan to focus on the segmentation of small lesions to further improve the performance of, and ultimately promote, the DR segmentation task to be widely used in clinical diagnosis.

## Figures and Tables

**Figure 1 diagnostics-12-02918-f001:**
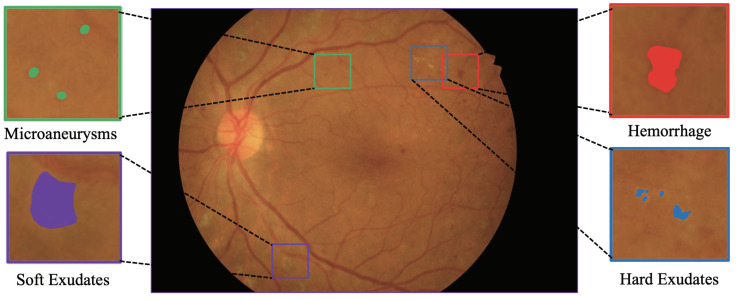
Illustration of fundus image with DR lesion.

**Figure 2 diagnostics-12-02918-f002:**
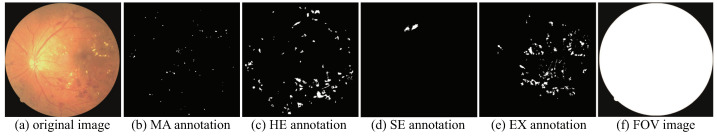
Illustration of fundus image, DR lesions, and FOV mask image.

**Figure 3 diagnostics-12-02918-f003:**
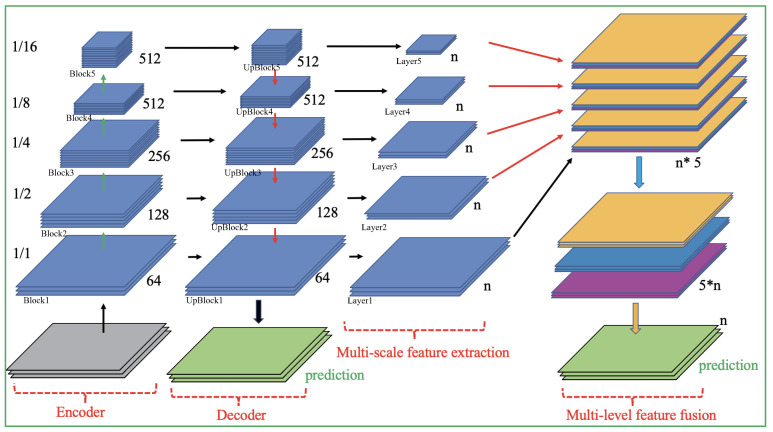
The ovaerall architecture of MSLF-Net. (Green arrow denotes down-sampling, red arrow denotes up-sampling, black arrow denotes information flow, blue arrow denotes channel rearrange, orange arrow denotes channel fusion, n = number of lesion types + 1).

**Figure 4 diagnostics-12-02918-f004:**
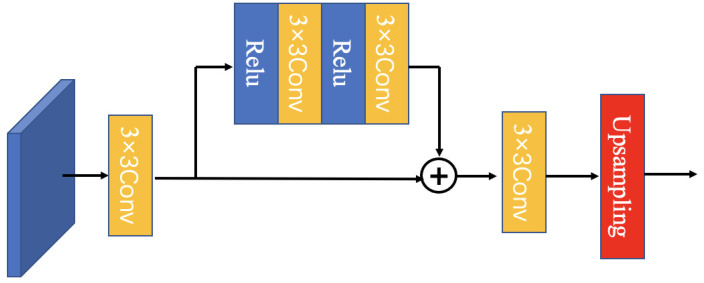
The stucture of the MSFE module.

**Figure 5 diagnostics-12-02918-f005:**
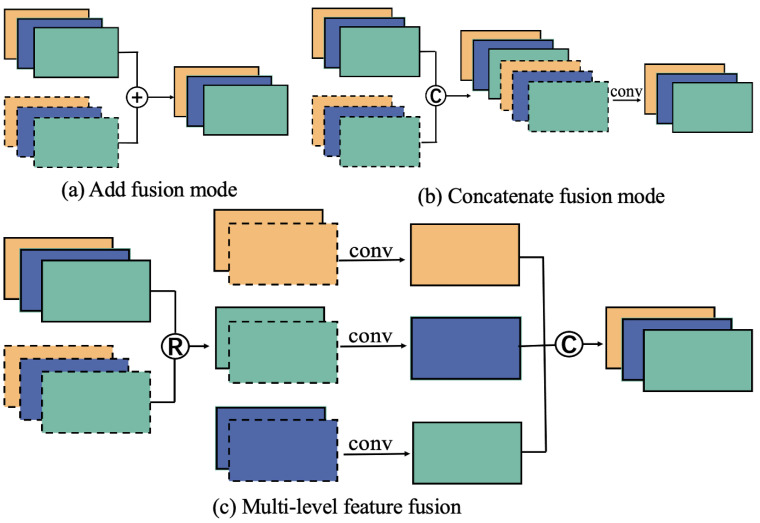
Detailed description of simple feature fusion and multi-level feature fusion.

**Figure 6 diagnostics-12-02918-f006:**
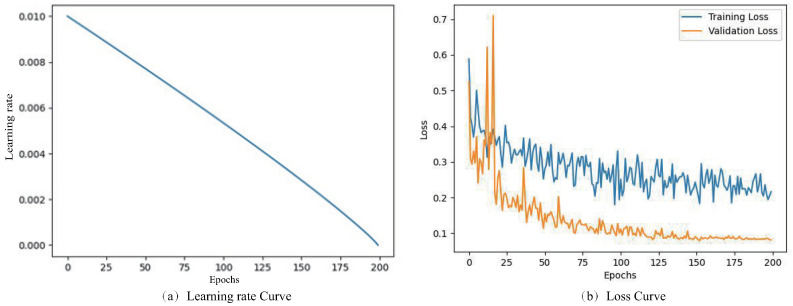
Learning rate curve and loss curve.

**Figure 7 diagnostics-12-02918-f007:**
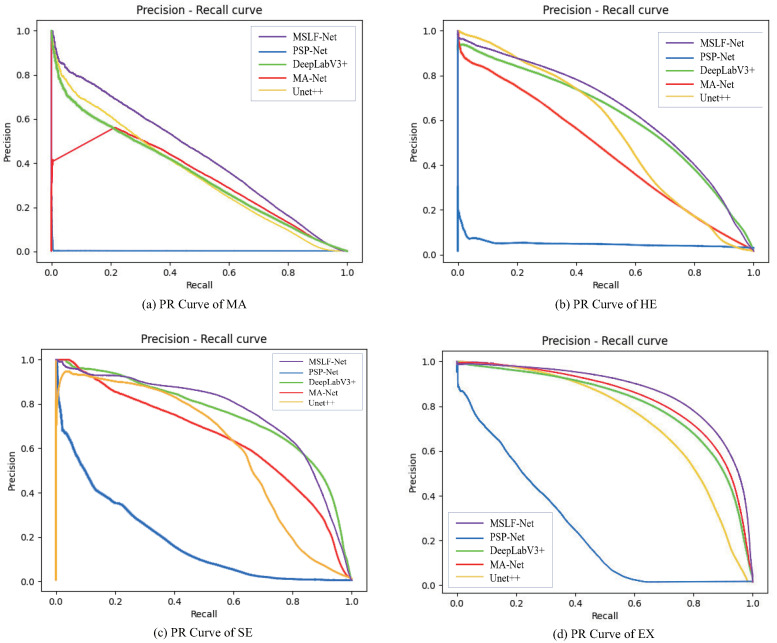
PR curves of MSLF-Net on IDRID.

**Figure 8 diagnostics-12-02918-f008:**
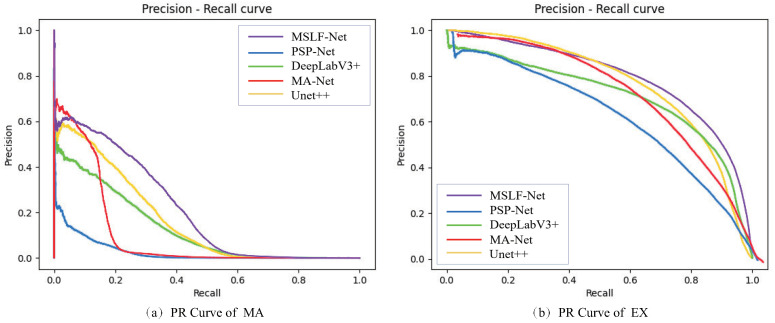
PR curves of MSLF-Net on e_ophtha.

**Figure 9 diagnostics-12-02918-f009:**
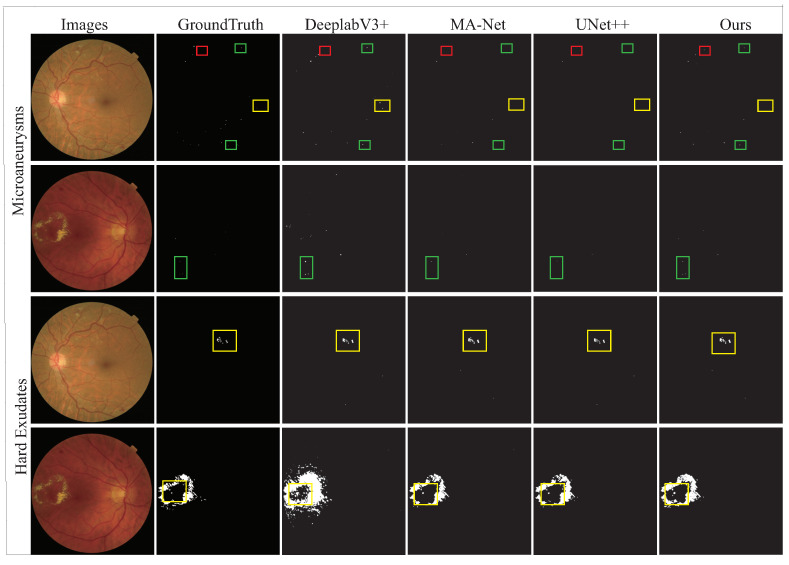
Illustration of fundus image with DR lesion. The rectangular box in the figure shows the comparison between the segmentation results of various methods and Ground Truth. The yellow boxes indicate that our method reduces the possibility of false detection compared with other methods, the green boxes indicate that our method reduces the possibility of false detection, while the red boxes indicate the error of our method.

**Figure 10 diagnostics-12-02918-f010:**
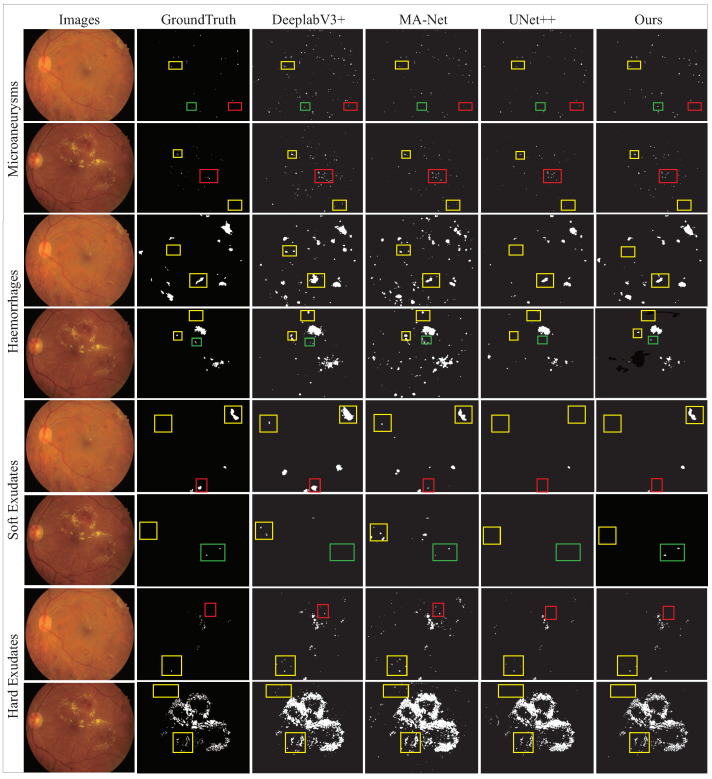
Illustration of fundus image with DR lesion.

**Table 1 diagnostics-12-02918-t001:** The detailed configuration of MSLF-Net.

Stage	Layers	Input	Input Size	Operations	Output Size
Encoder	Block1	Input image	1440 × 960 × 3	Conv Maxpool	720 × 480 × 64
	Block2	Block1	720 × 480 × 64	Conv Maxpool	360 × 240 × 128
	Block3	Block2	360 × 240 × 128	Conv Maxpool	180 × 120 × 256
	Block4	Block3	180 × 120 × 256	Conv Maxpool	90 × 60 × 512
	Block5	Block4	90 × 60 × 512	Conv Maxpool	90 × 60 × 512
Decoder	UpBlock5	Block5	90 × 60 × 512	Conv Upsample Concat	180 × 120 × 256
	UpBlock4	Block5 Block4	180 × 120 × 512	Concat Conv UpSample	360 × 240 × 128
	UpBlock3	Block4 Block3	360 × 240 × 256	Concat Conv UpSample	720 × 480 × 64
	UpBlock2	Block3 Block2	720 × 480 × 128	Concat Conv UpSample	1440 × 960 × 64
	UpBlock1	Block2 Block1	1440 × 960 × 64	Concat Conv UpSample	1440 × 960 × 5
MSFE	Layer5	UpBolck5	90 × 60 × 512	Conv UpSample	1440 × 960 × 5
	Layer4	UpBolck4	180 × 120 × 512	Conv UpSample	1440 × 960 × 5
	Layer3	UpBolck3	360 × 240 × 256	Conv UpSample	1440 × 960 × 5
	Layer2	UpBolck2	720 × 480 × 128	Conv UpSample	1440 × 960 × 5
	Layer1	UpBolck1	1440 × 960 × 64	Conv	1440 × 960 × 5
MLFF	Fusion	Layer 1-5	1440 × 960 × 25	Fusion	1440 × 960 × 5

**Table 2 diagnostics-12-02918-t002:** Comparison experiments based on IDRID.

Method	MA	HE	SE	EX	mAUC_PR
FRCN	0.3383	0.4200	0.5152	0.5472	0.4552
UNet++	0.3535	0.5472	0.6052	0.7415	0.5619
MANet	0.3220	0.4607	0.6466	0.8294	0.5647
PSPNet	0.0024	0.0340	0.1825	0.2483	0.1168
DeeplabV3+	0.3500	0.6142	0.7416	0.8042	0.6275
Paper [27]	0.3499	0.5273	0.6449	0.7229	0.5613
L-Seg	0.4627	0.6374	0.7113	0.7945	0.6515
RTNet-base	0.4279	0.6570	0.5968	0.8659	0.6369
MSLF-Net	0.4393	0.6411	0.7597	0.8644	0.6761

**Table 3 diagnostics-12-02918-t003:** Comparison experiments based on e_ophtha.

Method	MA	HE	SE	EX	mAUC_PR
FRCN	0.0269	-	-	0.1675	0.0972
UNet++	0.1636	-	-	0.7649	0.4643
MANet	0.1098	-	-	0.7216	0.4157
PSPNet	0.0237	-	-	0.6161	0.3199
DeeplabV3+	0.1281	-	-	0.7105	0.4193
Paper [27]	0.1367	-	-	0.6578	0.3973
L-Seg	0.1687	-	-	0.4171	0.2929
MSLF-Net	0.2179	-	-	0.8001	0.5090

**Table 4 diagnostics-12-02918-t004:** Performance comparison of different model components.

Method	MA	HE	SE	EX	mAUC_PR
Baseline	0.4059	0.6348	0.7017	0.8460	0.6471
+MSFE	0.3959	0.6446	0.7160	0.8464	0.6507
+MLFF	0.4393	0.6411	0.7597	0.8644	0.6761

**Table 5 diagnostics-12-02918-t005:** Performance comparison of weightage of hybrid loss function.

λ	MA	HE	SE	EX	mAUC_PR
−1	0.3826	0.6227	0.6635	0.8576	0.6316
0	0.4017	0.6174	0.7533	0.8457	0.6545
0.1	0.4234	0.6154	0.7490	0.8561	0.6604
0.2	0.4187	0.6162	0.7116	0.8091	0.6389
0.3	0.4325	0.6360	0.7300	0.8410	0.6599
0.4	0.4094	0.6179	0.7623	0.8469	0.6591
0.5	0.4198	0.6489	0.7324	0.8540	0.6638
0.6	0.4339	0.6290	0.7451	0.8517	0.6649
0.7	0.4227	0.6311	0.7508	0.8278	0.6581
0.8	0.3889	0.6431	0.7498	0.8419	0.6559
0.9	0.4459	0.6472	0.7394	0.8513	0.6709
1.0	0.4393	0.6411	0.7597	0.8644	0.6761
1.1	0.4469	0.6354	0.7211	0.8511	0.6636
1.2	0.4328	0.6390	0.7327	0.8479	0.6631

**Table 6 diagnostics-12-02918-t006:** Performance comparison of weightage of hybrid loss function.

Operation	MA	HE	SE	EX	mAUC_PR
baseline	0.3326	0.5568	0.6236	0.7594	0.5681
+Augmentation	0.3742	0.5754	0.6756	0.8394	0.6162
+FOV	0.4393	0.6411	0.7597	0.8644	0.6761

## Data Availability

We used two public datasets to evaluate our proposed segmentation model. They are the IDRID dataset and e_ophtha dataset. The two datasets can be accessed from the following URLs: https://ieee-dataport.org/open-access/indian-diabetic-retinopathy-image-dataset-idrid and https://www.adcis.net/en/third-party/e-ophtha (accessed on 28 August 2022). The mask images that we added to the dataset can be accessed from the following URLs: https://drive.google.com/file/d/1Gt5MHvo_0qVboXV155c9t0ZdBwHwpT1c/view?usp=sharing (accessed on 21 September 2022).

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
