# Peer review of "MSLF-Net: A Multi-Scale and Multi-Level Feature Fusion Net for Diabetic Retinopathy Segmentation"

_diagnostics, 2022, doi:10.3390/diagnostics12122918_

Round 1
Reviewer 1 Report
Authors need to address the following suggestons.
1. Authors need to justify how they arrive at the hybrid loss function. Why only 10% weightage is given for dice loss ?
2. Hyperparameter details need to be included.
3. Data augmentation details need to be included.
4. Multifeature fusion is already employed in the existing works Authors need to discuss how the proposed work is different from the other works. Few such works are mentioned here. Authors need to discuss the following works in the related works and show case their novelty.
DeepFusion: A deep learning based multi-scale feature fusion method for predicting drug-target interactions
A Novel Multi-Feature Fusion Method for Classification of Gastrointestinal Diseases Using Endoscopy Images
5. Comparsion with SOTA methods need to be included.
6. Novelty of the proposed work need to be highlighted clearly.
Reviewer 2 Report
Dear Authors,
The proposed work presents a CNN supported segmentation methodology to extract the abnormal section in Fundus Retinal image with Ground-Truth.
I request you to answer the following comments and improve the articles as per the suggestion (If applicable).
General Suggestions:
1. The introduction section is good. I suggest you to add a context section after the introduction to discuss the Similar CNN segmentation schemes implemented on the retinal image to detect the DR lesion segmentation.
2. Fig 2 (c) Must be Original image (instead images)
3. The term "We resize the pictures of the IDRID and e_ophtha to 1440*960 without deformation" must be verified with the image size represented in Encoder and decoder section. (Since, most of the CNN segmentation in literature considered 512x512x3 images, which is simple in implementation compared to the proposed dimension.
Specific Comment:
1. The general procedure in CNN segmentation is; Training the considered CNN scheme (Encoder-Decoder scheme) with the test image and the annotations. This demonstration and the discussion can be included in this work. Without the annotation or ground-truth image, the CNN will not learn about the section to be segmented.
2. Include a Graph to denote the training and validation accuracy as well as the loss function, which is common in CNN segmentation schemes.
3. For few chosen images, implement a relative analysis between the segmented section and the annotations and measure the achieved metrics.
4. Present a graphical comparative study to justify the merit of the presented work with existing methods.
5. Table 2 to 4 presents the achieved results. These are mean values or individual values? Please justify.
Please carefully improve the paper and include as the necessary information to justify the merit of the achieved results.
Round 2
Reviewer 2 Report
Dear Authors,
All my comments are addressed with appropriate results.
Thank you.